# The Role of Extra-Operative Cortical Stimulation and Mapping in the Surgical Management of Intracranial Gliomas

**DOI:** 10.3390/brainsci12111434

**Published:** 2022-10-25

**Authors:** Kostas N. Fountas, Alexandros Brotis, Thanasis Paschalis, Eftychia Kapsalaki

**Affiliations:** 1Department of Neurosurgery, Faculty of Medicine, University of Thessaly, 41200 Larisa, Greece; 2Department of Neurosurgery, General University Hospital of Larissa, 41200 Larissa, Greece; 3Department of Radiology, Faculty of Medicine, University of Thessaly, 41200 Larisa, Greece; 4Department of Neuroradiology, Institute Euromedica-Encephalos, 11528 Athens, Greece

**Keywords:** electrode, extra-operative, glioma, mapping, stimulation, subdural

## Abstract

Background: Aggressive resection without compromising the patient’s neurological status remains a significant challenge in treating intracranial gliomas. Our current study aims to evaluate the efficacy and safety of extra-operative stimulation and mapping via implanted subdural electrodes with or without depth (EOCSM), offering an alternative approach when awake mapping is contraindicated. Methods: Fifty-one patients undergoing EOCSM for glioma resection in our institution formed the sample study of our current retrospective study. We assessed the effectiveness and safety of our approach by measuring the extent of resection and recording the periprocedural complications, respectively. Results: The mean age of our participants was 58 years (±9.4 years). The lesion was usually located on the left side (80.4%) and affected the frontal lobe (51.0%). EOCSM was successful in 94.1% of patients. The stimulation and electrode implantation procedures lasted for a median of 2.0 h and 75 h, respectively. Stimulation-induced seizures and CSF leakage occurred in 13.7% and 5.9% of our cases. The mean extent of resection was 91.6%, whereas transient dysphasia occurred in 21.6% and transient hemiparesis in 5.9% of our patients, respectively. Conclusions: Extraoperative stimulation and mapping constitute a valid alternative mapping option in glioma patients who cannot undergo an awake craniotomy.

## 1. Introduction

There is a growing body of evidence identifying the importance of radical resection for intracranial gliomas, not only regarding progression-free and overall survival but also in terms of life quality and perioperative complications [1,2,3,4,5,6,7,8,9]. Many highly sophisticated imaging and electrophysiological modalities aim to increase safety during aggressive surgical resection. Intraoperative mapping with direct electrical cortical stimulation (DCS) remains the most accurate method to identify the eloquent cortical and subcortical areas [10].

The employment of DCS requires a cooperative, emotionally stable patient to undergo an awake craniotomy [11]. Although most patients can go through an awake craniotomy, some patients are reluctant or unable to go through such a process [11,12,13,14]. Modalities such as task-generated or resting-state functional MRI (fMRI) and transcranial magnetic stimulation (TMS) may be used in these circumstances but with highly variable reproducibility and accuracy rates [15,16,17,18,19,20,21,22]. Of note, the accuracy rates of fMRI and TMS in identifying speech-associated cortical areas have been reported as low as 79% and 8%, respectively [15,16,17,18,19,20,21,22].

Alternatively, extra-operative cortical stimulation and mapping (EOCSM) through previously inserted subdural electrodes with or without depth electrodes may constitute a valid alternative for accomplishing a maximal glioma resection without compromising the patient’s neurological status [10,11]. Our current study presents our experience in employing EOCSM in a series of patients with intracranial gliomas. Our primary aim was to assess the efficacy and safety of the EOCSM, while our secondary objective was to identify potential outcome modifiers.

## 2. Methodology

### 2.1. Study Design

Our current study constitutes a retrospective analysis of prospectively collected anonymized data covering 13 years (2007–2019). The retrospective nature of our study, which was based on anonymized hospital data, exempted our non-experimental study from our hospital’s Institutional Review Board (IRB) approval. In addition, we handled all patient data according to the World Medical Association Declaration of Helsinki and the current Health Insurance Portability and Accountability Act (HIPAA) regulations [23]. Equally important, we obtained written consent from all participants before each surgical procedure.

### 2.2. Eligibility Criteria

The inclusion criteria in our study were: (i) adult patients (≥17 y.o.) with magnetic resonance imaging (MRI) presumptive diagnoses of glioma in or adjacent to eloquent cortical areas, (ii) patients not willing to undergo an awake craniotomy, (iii) patients with an absolute or a relative contraindication for undergoing an awake craniotomy (severely obese patients, patients with pre-existing respiratory disease, heavy smokers, patients with documented sleep apnea, emotionally unstable patients), and (iv) patients with a long history of tumour-related seizures. Our exclusion criteria included patients’ refusal to participate in our study, pediatric patients (<17 y.o.), and patients with a previous craniotomy.

### 2.3. Preoperative Planning

All preoperative MRI studies were performed on a 3T MRI scanner (GE Healthcare, Milwaukee, WI, USA) and included 3D T1 images, axial T2, 3D fluid-attenuated inverse recovery (FLAIR), diffusion weighted images (DWI), T2*, and MR venography sequences, as well as post-contrast 3D T1 images. In selected cases, we obtained single-voxel proton MR spectroscopy (^1^HMRS), task-generated functional MRI, and Fractional Anisotropy/Diffusion Tensor Imaging (FA/DTI). Our preoperative planning was based on the anatomic location of the lesion and the anticipated eloquent areas according to all imaging and EEG data. The surgical plan covered all these areas with the appropriate combination of subdural strips and grid electrodes, as well as depth electrodes, if necessary. The principal author (KNF) performed the procedure in three stages, including a craniotomy for electrode implantation, an extra-operative stimulation and mapping phase, and a tumour resection phase.

### 2.4. Electrode Implantation

We performed the implantation craniotomy under general anaesthesia with the head fixed on a three-point Mayfiled head clamp. An inverse question mark skin incision permitted a broad fronto-parieto-temporal craniectomy. After making an adequate dural opening, we inserted the preselected subdural and depth electrodes, implanted the grid/strip electrodes under direct vision to avoid any venous injuries, and secured them onto the adjacent dura (Figure 1). The electrode tails were tunnelled under the skin, exiting 3–4 cm away from the skin incision to mitigate the risk of infection, and we safely secured them to the skin with a figure-8 suture. A bone groove at the craniotomy edges minimized mechanical friction and damage to the electrodes. Collodion was applied at the skin exit point to prevent any cerebrospinal fluid (CSF) leakage. The day after surgery, the patient obtained a head computed tomography (CT) and a brain MRI scan to verify the final position of the implanted electrodes (Figure 2).

### 2.5. Stimulation and Mapping Phase

Stimulation was routinely performed at least 24 h after electrode implantation in the patient’s room, with the patient wide awake and in a comfortable position. The stimulation parameters were 50 Hz frequency, 3 s train, 0.2 ms alternating polarity square-wave pulses, with current amplitude 2–4 mAmps using a Grass 88 stimulator. In the case of a positive result, we performed three rounds of stimulation to verify the results. The language testing included a minimum of four tests: spontaneous speech, Boston naming test, reading, and listening comprehension. Other language tests were also employed depending on the patient’s linguistic skills and educational level. The whole procedure was completed in one or more sessions depending on the patient’s cooperation, the extent of stimulation and mapping, as well as the occurrence of any stimulation-induced seizure activity. Emotional, cognitive, or other autonomic responses were carefully evaluated and recorded. All stimulation and mapping data were registered on the obtained post-implantation CT and MRI studies, uploaded to the neuro-navigation workstation (StealthStation Surgical Navigation System, Medtronic, Minneapolis, MN, USA), and fused. A detailed cortical and subcortical functional map was created.

### 2.6. Tumour Resection Phase

The resection phase started by opening the previous skin incision and osseous flap. The main goal was to remove the implanted electrodes, the glioma, and other cortical areas associated with any epileptiform activity. However, we intended to avoid cortical and subcortical eloquent areas as identified by extra-operative stimulation (resection no closer than 10 mm). We routinely employed the subpial resection technique in all our cases, with minimal use of bipolar coagulation to avoid ischemic injuries.

### 2.7. Data Extraction

For our study two authors (AB and TP) gathered the hospital medical files, the patient’s preoperative and postoperative brain MRI studies, including fMRI and diffusion tensor imaging (DTI) sequences, and the operative report. Subsequently, we extracted the following data: (i) the patient’s demographic characteristics, (ii) the tumour location and histological grade, (iii) the extra-operative stimulation parameters and responses, (iv) the extent of resection, (v) the perioperative complications, if any, and (vi) the stimulation-associated complications whenever recorded. We used the latest World Health Organization (WHO) classification for the histological grading of the resected tumour [24,25]. An experienced neuroradiologist (EK), blinded to the surgeon’s perspective regarding the extent of resection, evaluated the postoperative MRI, obtained within 48 h after surgery, to estimate the actual extent of resection. In the case of high-grade gliomas, the extent of resection was based on the contrast-enhancing volume. In contrast, FLAIR images were utilized in low-grade gliomas. Resection of ≥ 95% of the tumour was considered gross total resection (GTR).

### 2.8. Statistical Analysis

We used parametric and non-parametric statistics to summarize the basic characteristics of our study sample. The search for predictors utilized univariate and multivariate logistic regression of univariate models and recursive partitioning. The effect size estimate was described in odds ratio (OR) along with its 95% confidence interval (95% CI). All statistical analyses were performed using an R statistical environment [26].

## 3. Results

### 3.1. Study Participants

A total of 536 glioma cases were surgically treated during the study period. Among them, 54 patients (10.1%) were candidates for EOCSM via implanted subdural strip/grid and depth electrodes (Table 1). Finally, 51 patients (9.5%) underwent a two-stage extra-operative stimulation as well as mapping and resection of the underlying glioma. The mean age of our participants was 58.0 years (±9.4 years). Most of our cases (82.3%) were high-grade gliomas. The underlying glioma was in the left hemisphere in most of our participants (80.4%), with the frontal lobe being affected in 26 participants (51.0%). All participants underwent a conventional MRI study, while a task-generated fMRI study was available for 44 patients (86.3%), DTI/FA for 32 patients (62.7%), and^1^HMRS for 38 patients (74.5%).

### 3.2. Efficacy of EOCSM

The mean of stimulation sessions 1.4 (range:0–5, interquartile range: 1–2) corresponded to each patient. The stimulation and mapping procedure lasted for an average of 2 h (range: 0–4.5 h, interquartile range: 1.5–2.5 h). The mean electrode implantation time was 75.3 h (ranging from 48 to 120 h, interquartile range: 48–96). The extra-operative stimulation and mapping procedures were successful in most of our cases (94.1%), while in three patients (5.9%), the mapping could not be completed due to multiple stimulation-induced seizures. The average blood loss during the glioma resection procedure was 286 mL (range: 100–550 mL). The mean extent of glioma resection was 91.6% (range: 70–100%).

### 3.3. Complications Associated with Electrode Implantation and Stimulation

Seizures were encountered in seven patients (13.7%), followed by CSF leakage in three patients (5.9%). Of note, we observed a small-sized epidural hematoma, occasionally with a subdural extension, during the second procedure for removing the previously implanted electrodes in all cases (Figure 3). None of these hematomas, however, required surgical evacuation during the monitoring and the stimulation periods. There were no infections in our current series, although, in four patients (7.8%), the routinely obtained cultures from the explanted electrodes were positive. None of our patients developed any clinically significant edema, while persistent headache was encountered in six patients (11.8%), which was effectively treated with non-steroid, anti-inflammatory medications.

### 3.4. Complications Associated with Glioma Resection

The most common postoperative complication was transient dysphasia, occurring in 11 patients (21.6%), followed by transient hemiparesis in three patients (5.9%), subdural hematoma requiring urgent surgical evacuation in two patients (3.9%), and postoperative seizure activity in one patient (2.0%). The transient neurological symptomatology gradually resolved within three months. Our data analysis demonstrated that the anatomic location of the tumour in the temporal lobe (OR; 0.011; 95% CI: 0.005–0.073) was associated with a low probability of GTR (Figure 4, Table 2 and Table 3). Similarly, GTR was associated with increased intraoperative blood loss (OR: 1.02; 95% CI: 1.00–1.028), particularly in high-grade gliomas (Figure 5, Table 2). The present study identified no predisposing factors for the development of postoperative complications (Table 4).

## 4. Discussion

### 4.1. Overview of Our Findings

Our current study showed that EOCSM is an effective and safe alternative to awake glioma resection. Indeed, we achieved maximal glioma resection using EOCSM in 92% of our participants. At the same time, our complications included seizures and CSF leakage associated with the electrode implantation and stimulation, occurring in as high as 13.7% and 5.9% of our patients, respectively. The most frequently observed postoperative complication associated with the tumour resection phase were transient dysphasia and transient hemiparesis in as high as 21.6% and 5.9% of our cases, respectively.

### 4.2. Comparison of EOCSM with Other Non-Invasive Modalities

Several clinical investigators postulate that the employment of the resting-state fMRI may well increase the accuracy of non-invasive language mapping; however, there is not enough experience and clinical data to prove such a statement [18,19,20,21]. Similarly, the employment of TMS may well localize the sensorimotor cortex, but the accuracy in identifying speech-associated cortical areas remains questionable [22]. Although the reported sensitivity is 100%, the positive predictive value is as low as 34%, while the accuracy is only eight percent. It should be kept in mind that this is a rapidly evolving method, with a constantly increasing accuracy and clinical utilization [21,27,28,29,30,31,32].

Several authors studied the role of non-invasive electrophysiological and metabolic imaging methods in cortical mapping. The former included high-density electroencephalography (via 256 or 512 contacts), either alone or in conjunction with magnetic source imaging (MSI). From the metabolic imaging methods, positron emission tomography superimposed on MRI (PET-MRI) was the most extensively studied modality. However, the low accuracy rates of all these methods limit their broad use [11,12,33].

Moreover, the potential combination of all methods is expected to increase exponentially within the next few years, thus significantly improving their accuracy [33,34,35,36]. Lately, new reports have utilized artificial intelligence by combining the tools mentioned above and implementing the results on a patient’s MRI to maximize the surgeon’s armamentarium [37,38,39,40,41]. Despite the recent advances in MR-based techniques (fMRI, DTI/FA), and the non-invasive electrophysiological modalities (TMS, high-density EEG, MSI) for cortical and subcortical mapping, direct electrical stimulation and mapping remain the most accurate and reproducible modality for outlining functional areas, especially those associated with language production and comprehension [15,16,17,18,19,20,21,22].

The role of all these non-invasive tests in cortical mapping may well become more important shortly. However, these non-invasive tests cannot be used alone to accurately localize speech-associated cortical and subcortical areas. They can be employed to guide the subsequent direct electrical stimulation by identifying cortical and subcortical areas requiring further electrophysiological exploration for accurate mapping, thus accomplishing a safer and more radical glioma resection. Each of these non-invasive methods may be used in association with DCS, primarily to decrease the length of the stimulation process.

### 4.3. Comparison of EOCSM with DCS

The identification and accurate delineation of white matter tracts is of great importance. We use preoperative DTI for identifying the tracts at risk each time, and then depth electrodes may be inserted along their pathway for stimulating and mapping in detail these tracts. However, it is beyond any doubt that intraoperative stimulation is superior to extra-operative stimulation and mapping through multiple depth electrodes or stereo-EEG [42,43,44,45,46,47,48,49,50,51].

Although DCS constitutes the gold standard for cortical mapping, it requires a cooperative patient and a specially trained neuroanesthesiologist, familiar with awake procedures. Severe obesity, a history of heavy smoking, a history of a pre-existent respiratory disease, a history of sleep apnea, a patient older than 65 years (an age limit widely accepted although it may vary from one institution to another, while sporadic cases in elderly patients have been published [13]), and elicitation of cardio-trigeminal reflex have been reported as absolute or relative contraindications for an awake craniotomy. Moreover, an emotionally unstable or fearful patient may not be willing to undergo and may not consent to undergo an awake craniotomy. The previous series reported that the failure rate of an awake craniotomy ranged between 0.5% and6.3% [12,14]. It also needs to be taken into consideration that special age groups such as pediatric and elderly patients may generally not be candidates for an awake procedure. In all these cases, where DCS is necessary for more aggressive tumour resection and an awake procedure cannot be safely performed, EOCSM via implanted subdural with or without depth electrodes represent a valid alternative option.

The advantages of EOCSM represent the disadvantages of DCS through an awake craniotomy, and vice versa. Indeed, the major advantage of extra-operative stimulation is its applicability among pediatric and elderly patients, who generally are not good candidates for an awake procedure. Similarly, non-cooperative and emotionally unstable patients may benefit from extra-operative cortical stimulation and mapping, as may multi-lingual patients, in whom mapping of all language-associated cortical and subcortical areas may not be feasible during an awake procedure due to time limitations. Another advantage of the extra-operative stimulation and mapping method, except in a stressed patient, is recording various emotional, neurocognitive, and autonomic responses in a relatively uncomfortable position, in the operating room. Contrariwise, the risk of any procedure-associated complications is theoretically higher during the two surgical procedures of the extra-operative stimulation and mapping.

Additionally, the cost of two surgical procedures, along with the cost of the implanted electrodes, may be higher than that of a single but lengthier, awake craniotomy. Furthermore, EOCSM can be performed via the implanted depth electrodes; therefore, it is significantly more limited than an open, awake procedure. It must be emphasized that many of the complications and difficulties associated with EOCSM may be further limited in the near future, since many technological advances may well increase the accuracy of this method and minimize its adverse events. The development of hybrid electrodes with microwire arrays embedded between the contacts of the subdural grid electrodes may further increase the accuracy of the performed stimulation and mapping. Also, the designing of special strip electrodes for covering specific anatomical areas such as the insula, as well as manufacturing of high-density subdural grid electrodes, may further increase the accuracy of the stimulation and mapping. Moreover, the development of newer, ier biomaterials that are friendlier than the traditionally used silicone may further decrease electrode-associated complications in the near future.

### 4.4. Procedure Technical Tips & Tricks

Extra-operative stimulation and mapping require two surgical procedures, one for implantation and the second for electrode removal and resection of the underlying glioma. This increases the possibility of any procedure-associated complications. All non-invasive, preoperative tests, such as fMRI, FA/DTI, TMS, MEG/MSI, and surface EEG, may be used to guide implantation of the electrodes and to rationalize their proper shape and contact number selection for covering all the areas of interest. Therefore, it is of paramount importance that both procedures are planned and executed as meticulously as possible for mitigating the chance of any adverse events or complications. 

Obtaining a preoperative CT or MR venography may provide significant information regarding the cortical venous anatomy, enabling safer and more efficacious implantation surgical planning. We also employed preoperative DTI in our patients (as we routinely do ), mostly for guiding us in the gross location of the tracts at jeopardy each time. This may facilitate the whole stimulation process and diminish the necessary time for stimulation and mapping; at the same time the accuracy of the DTI can be assessed intraoperatively by the employed stimulation [42,43,44,45,46,47,48,49,50,51]. An adequate-size craniotomy for implanting all the electrodes under direct vision to avoid any surface venous injuries is essential. Securing the implanted electrode is also greatly important for avoiding any electrode migration during the stimulation and mapping, which may affect the accuracy and the clinical value of the obtained information. Likewise, bone grooving at the craniotomy edges can minimize mechanical friction of the electrode tail, thus minimizing the chance of electrode dislocation or breakage. Water-tight dural closure is important for avoiding any post-implantation CSF leakage and for minimizing the risk of an infection, but also for avoiding any brain parenchyma shift and electrode migration with an obvious effect on the accuracy of the performed mapping. Another technical consideration, especially when many bulky subdural grid electrodes are used, is the administration of steroids to decrease the incidence of any brain edema, particularly in pediatric patients. The development of edema may cause an increase in intracranial pressure, which may be catastrophic, especially in a patient with a large glioma. In cases in whom edema development is anticipated, monitoring of the intracranial pressure is recommended during the entire implantation period. The administration of systemic antibiotics during implantation remains controversial in the pertinent literature. We prefer to cover all our patients with antibiotic prophylaxis during the entire implantation period. We encountered no infections in our current series, with the mean length of implantation approximating three days (75.3 h).

### 4.5. Prognostic Factors

The anatomic location of the glioma was associated with the extent of resection in our current series. As expected, a glioma in the left temporal lobe had the lowest probability of GTR. This can be explained by the more frequent involvement of the left (dominant) temporal lobe in speech, which limited the extent of resection, despite the employment of cortical stimulation and mapping. Furthermore, increased intra-operative blood loss, which may indicate more aggressive resection, was associated with a greater extent of resection in our series, especially among cases of high-grade gliomas.

Our data analysis identified no predisposing factors for the development of perioperative complications. Thorough knowledge of procedure-associated and stimulation-induced complications is crucial for carefully avoiding them and adequately managing them when they occur. The development of edema has been reported to vary from 0.5% to 14%, with a higher chance when large, interhemispheric subdural grids are implanted [52,53,54,55,56,57,58,59]. We experienced no problems with increased ICP in our current series. The development of post-implantation epidural hematoma has been reported to be from 1.8% to2.5%, while the incidence of subdural hematomas has been reported to range from 1.1% to 14% [52,53,54,55,56,57,58,59]. In our series, we had no incidents of epidural or subdural hematoma requiring surgical intervention. It must be mentioned, however, that the vast majority of our patients had a small epidural blood clot with or without subdural extension, without any clinical significance. The risk of infection has been reported to be from 1.1% to17% among previously published series [52,53,54,55,60,61]. We had no infected cases, although 7.8% of our participants had positive cultures, which were treated with antibiotics with no other consequences. The incidence of CSF leakage has been reported to be from 0% to20% [52,53]. We encountered CSF leakage in 5.9% of our patients. The most commonly occurring adverse event during the stimulation and mapping process is the development of seizures. It has been reported to be as high as 33% [52,53]. We experienced seizure-related problems in 13.7% of our cases, which was the underlying cause of all our failed cases (5.9%). The management of seizures can be safely done with the intravenous administration of anticonvulsants. Higher current amplitude may be implicated in a higher incidence of stimulation-induced seizures. Initiating the stimulation process from less epileptogenic (distant from the pathology) areas may facilitate the whole process and thus minimize the risk of failure.

### 4.6. Study Limitations

Our current study carries many weaknesses and limitations. First, it carries all the limitations and biases of a retrospective study. Second, it reports a single institution’s experience based on a relatively small number of cases. Moreover, it must be taken into consideration that a single neuroradiologist calculated the extent of resection. Therefore, any inter-observational variations cannot be ruled out. Another limitation is that our study was based on hospital data files. In these data files, higher cognitive functions, including neuropsychological testing, and the patient’s quality of life were not systematically recorded, and were, thus, not included in our analysis. Moreover, our current case series lacks a comparator arm with patients undergoing awake glioma resection. The baseline characteristics of the two groups were not comparable, due to inherent differences in the treatment indications. Therefore, such a comparison was not carried out to avoid allocation bias and the effect of the potential presence of several confounders. All these parameters need to be taken into account for extracting more accurate conclusions. Larger, multi-institutional, prospective clinical studies are necessary for further exploring and accurately outlining the role of extra-operative stimulation and mapping in the surgical management of patients with intracranial gliomas, including the overall cost of this method, as well as its potential to provide valuable research to cortical stimulation data. Equally important is the future prospective need to study the role of EOCSM on higher cortical functions and overall quality of life.

## 5. Conclusions

Many strategies have been developed for maximal but safe glioma resection. Among them, direct cortical stimulation and mapping via an awake craniotomy are generally considered the most accurate and efficacious methodology in maximizing tumour resection, without jeopardizing vital neurological functions. In cases where an awake procedure cannot be performed, extra-operative cortical stimulation via implanted subdural electrodes with or without depth may be an alternative mapping option. Thorough knowledge of this technique, its tips and tricks, and its possible complications can provide a valuable tool in the surgical management of glioma patients. This method may be complementary to direct cortical stimulation and mapping.

## Figures and Tables

**Figure 1 brainsci-12-01434-f001:**
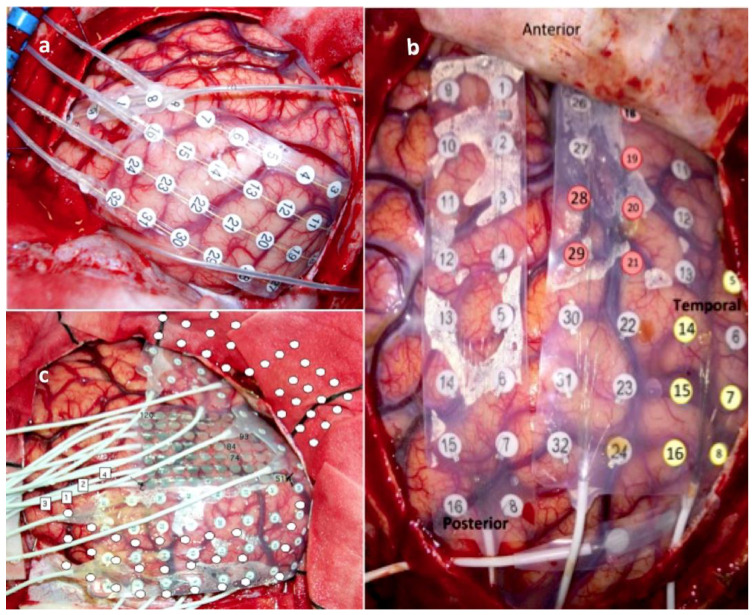
(**a**–**c**) Intraoperative photos (during the implantation procedure) demonstrate various combinations of strip and grid subdural electrodes for extra-operative stimulation and mapping. Adequate cortical exposure is required for safer electrode implantation.

**Figure 2 brainsci-12-01434-f002:**
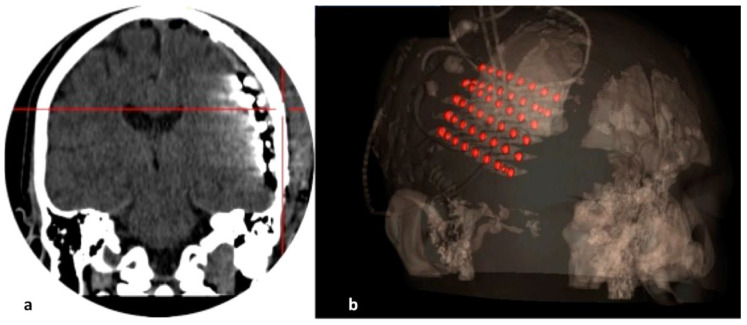
(**a**,**b**) Postimplantation CT scan (coronal view) and 3D model are used for verifying the location of the implanted electrodes and their contacts.

**Figure 3 brainsci-12-01434-f003:**
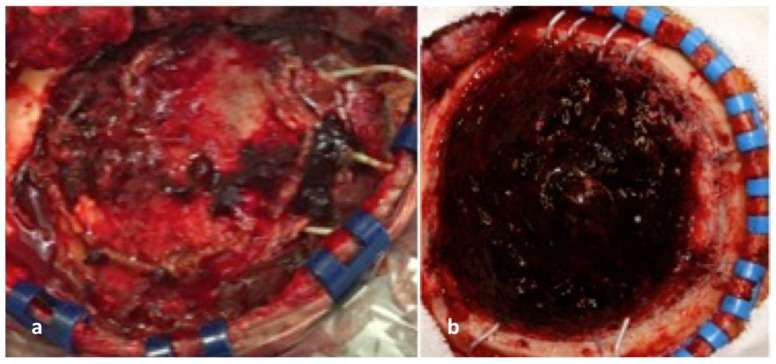
(**a**,**b**) Intra-operative photos (during the electrode removal and the tumour resection procedure) depict small-size epidural hematomas, which required no surgical evacuation and had no clinical significance.

**Figure 4 brainsci-12-01434-f004:**
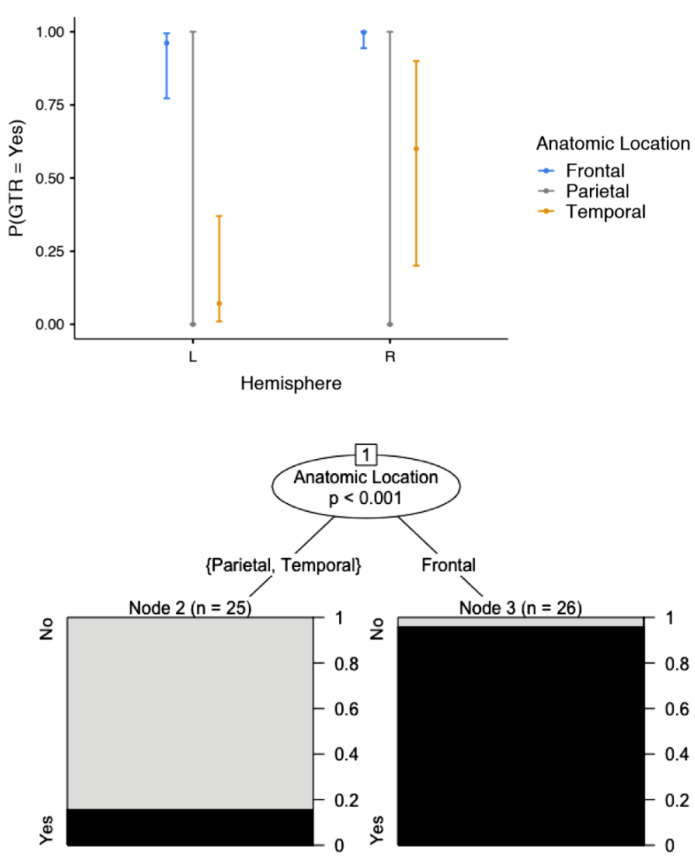
The extent of resection was influenced by the anatomic location of the glioma.

**Figure 5 brainsci-12-01434-f005:**
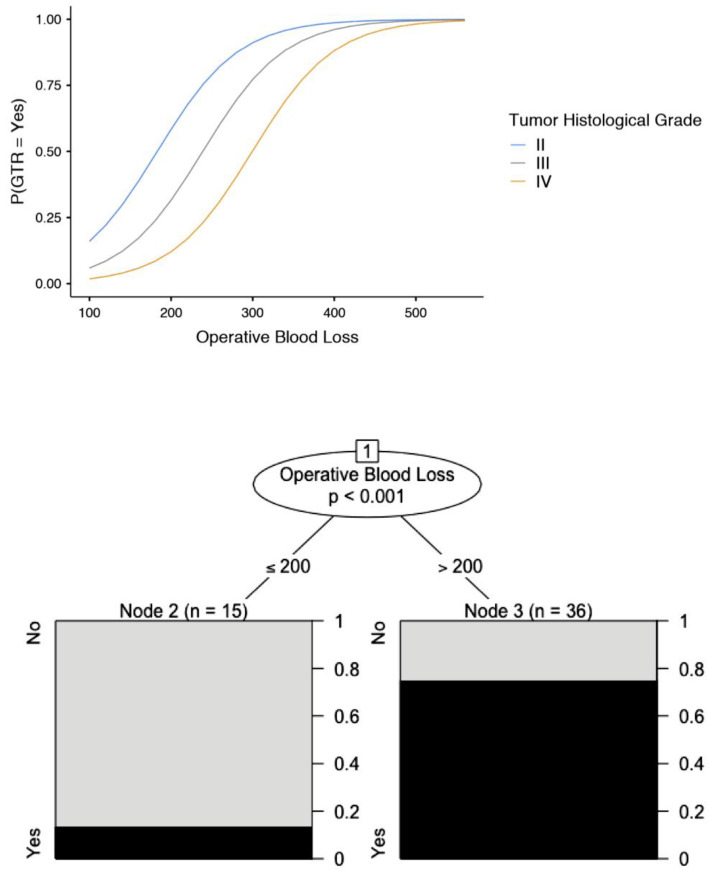
The intra-operative blood loss was positively affected by the histological grade of the glioma and the extent of resection.

**Table 1 brainsci-12-01434-t001:** This is a summary of basic characteristics of our study sample (N = 51).

		Mean or Median *	SD or IQR *	*p* (Shapiro-Wilk Test)
Age (years)		58	9.38	0.22
Number of stimulation sessions		1 *	1–2 *	<0.001
Cumulative duration of stimulation (h)		2 *	1.5–2.5 *	0.009
Cumulative implantation time (h)		72 *	48–96 *	<0.001
Intraoperative blood loss (mL)		286	96	0.157
Extent of resection (%)		95	85–98	<0.001
		Counts	%	*p* (Chi-square)
Gender	Female	21	41.2	0.262
	Male	30	58.8	
Availability of ^1^HMRS	No	13	25.5	<0.001
	Yes	38	74.5	
Availability of fMRI	No	7	13.7	<0.001
	Yes	44	86.3	
Availability of DTI/FA	No	19	37.3	0.092
	Yes	32	62.7	
Location (hemisphere)	Left	41	80.4	<0.001
	Right	10	19.6	
Location (lobe)	Frontal	26	51	0.002
	Parietal	6	11.8	
	Temporal	19	37.2	
Stimulation associated complications	No	41	80.4	<0.001
	CSF leak	3	5.9	
	Seizures	7	13.7	
Perioperative complications	No	34	67.7	0.024
	Transient Dysphasia	11	21.6	
	Transient Hemiparesis	3	5.9	
	SDH	2	3.9	
	Seizures	1	2.1	
WHO tumour histological grade	II	9	17.6	<0.001
	III	15	29.4	
	IV	27	52.9	
GTR	No	22	43.1	0.327
	Yes	29	56.9	

The parameter was described using the median and interquartile range due to deviation from the normal distribution. (SD, standard deviation; IQR, interquartile range; ^1^HMRS, MR spectroscopy; fMRI, functional MRI; DTI/FA; diffusion tensor imaging and fractional anisotropy; GTR, gross total resection).

**Table 2 brainsci-12-01434-t002:** The univariate logistic regression analysis according to generalized linear models showed that the location of the tumor (lobe) and the total intra-operative blood loss were predictors for tumour GTR.

Risk Factor	Reference	Comparator	Univariate Analysis	
			Crude OR (95% CI)	*p*
Age (years)	Per year	(-)	0.962 (0.899–1.02)	0.219
Gender	Female	Male	0.71 (0389–1.25)	0.234
Availability of ^1^HMRS	No	Yes	0.879 (0.448–1.66)	0.694
Availability of fMRI	No	Yes	1.993 (0.872–5.47)	0.103
Availability of DTI/FA	No	Yes	1.15 (0.644–2.042)	0.639
Location (hemisphere)	Left	Right	0.497 (0.218–1.17)	0.067
Location (lobe)	Frontal	Parietal	0 (0-Infinity)	0.993
		Temporal	0.011 (0.005–0.073)	<0.001
Number of stimulation sessions	Per stimulation	(-)	1.21 (0.656–2.46)	0.562
Cumulative duration of stimulation	Per hour of stimulation	(-)	1.40 (0.822–2.53)	0.236
Stimulation associated complications	No	Yes	0.497 (0.218–1.17)	0.067
Cumulative implantation time	Per hour of stimulation	(-)	1.00 (.0981–1.03)	0.780
Intraoperative blood loss	Per mL of blood lost	(-)	1.02 (1.00–1.028)	0.001
WHO tumor grade	II	III	0.429 (0.05–2.567)	0.377
		IV	0.265 (0.035–1.337)	0.136

(OR, odds ratio; ^1^HMRS, MR spectroscopy; fMRI, functional MRI; DTI/FA, diffusion tensor imaging and fractional anisotropy; GTR, gross total resection).

**Table 3 brainsci-12-01434-t003:** The multivariate regression analysis showed that the probability for gross total resection was limited in the left-sided temporal gliomas.

Risk Factor	Reference	Comparator	Multivariate Analysis	
			Adjusted OR (95% CI)	*p*
Location (hemisphere)	Left	Right	19.5 (1.30–293)	0.032
Location (Lobe)	Frontal	Parietal	0 (0-Infinity)	0.997
		Temporal	0.00308 (0.0–0.553)	<0.001

**Table 4 brainsci-12-01434-t004:** The univariate logistic regression analysis identified no predisposing factors for perioperative complication development.

Risk Factor	Reference	Comparator	Univariate Analysis	
			Crude OR (95% CI)	*p*
Age (years)	Per year	(-)	1.026 (0.963–1.098)	0.433
Gender	Female	Male	1.461 (0.799–2.818)	0.232
Availability of ^1^HMRS	No	Yes	0.859 (0.448–1.697)	0.650
Availability of fMRI	No	Yes	0.561 (0.234–1.27)	0.165
Availability of DTI/FA	No	Yes	0.609 (0.329–1.107)	0.106
Location (hemisphere)	Left	Right	2.023 (0.994–4.331)	0.055
Location (Lobe)	Frontal	Parietal	3.333 (0.504–22.71)	0.2
		Temporal	2.424 (0.675–9.179)	0.178
Number of stimulation sessions	Per stimulation	(-)	0.568 (0.201–1.172)	0.187
Cumulative duration of stimulation	Per hour of stimulation	(-)	0.859 (0.476–1.482)	0.593
Stimulation associated Complications	No	Yes	1.555 (0.759–3.202)	0.220
Cumulative Implantation Time	Per hour of stimulation	(-)	0.980 (0.951–1.005)	0.133
Intraoperative blood loss	Per mL of blood lost	(-)	0.998 (0.991–1.004)	0.497
WHO tumour grade	II	III	0.625 (0.11–3.404)	0.587
		IV	0.526 (1.108–2.60)	0.418
GTR	No	Yes	0.618 (0.333–1.115)	0.114

(OR, odds ratio; ^1^HMRS, MR spectroscopy; fMRI, functional MRI; DTI/FA, diffusion tensor imaging and fractional anisotropy; GTR, gross total resection).

## Data Availability

Not applicable.

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
