# Peer review of "The Role of Extra-Operative Cortical Stimulation and Mapping in the Surgical Management of Intracranial Gliomas"

_brainsci, 2022, doi:10.3390/brainsci12111434_

Round 1
Reviewer 1 Report
The present study about the value and role of EOCSM in brain glioma management/resection is well written and conducted; although methods sound and results are relatively interesting few problems are encountered as follow:
Minor:
1) All study sections are too long and redundant thus need to be systematically shortened (please amend).
2) It is not clear how this study, using anonymized data, does not need IRB approval (please clarify, explain, add details) and also who approved the patient's informed consent (please specify add details)?
Major:
1) Have the authors evaluated previous and after surgery superior cognitive function performances (neuropsychology testing) and if yes what was the impact of EOCSM on the latter (please add details)?
2) The study could be greatly improved if the authors would have a control group of similar brain glioma managed by awake surgery to compare the impact of such technique on patient’s final outcome.
3) Has any quality of life evaluation been done (please add details)?
4) Do the authors rely on DTI (please explain your own policy/phylosophy, add details)?
Reviewer 2 Report
This paper is well written. Aim was to assess the efficacy and safety of the EOCSM (extraoperative stimulation and mapping via implanted subdural electrodes) and meantime to identify potential outcome modifiers. Study design and statistical analysis are correct. Limitations of the study are well described, and conclusions are almost pertinent. However, the importance of preserving subcortical fibertracts in glioma resection is nowadays considered one of the upmost relevant factors; in my opinion, the Authors have to deserve at least a brief discussion to the argument, eventually adding some reference about preoperative and intraoperative technical methodologies actually reported in literature to implement safety and effectiveness in glioma surgery.
(simply suggested references are attached).

Round 2
Reviewer 1 Report
The authors answered satisfactorily to the reviewers criticisms although in this way they added more limitations to their study. The paper is globally improved.